# Application of Fuzzy TRUST CRADIS Method for Selection of Sustainable Suppliers in Agribusiness

Adis Puška [1,*], Miroslav Nedeljković [2], Ilija Stojanović [3] and Darko Božanić [4]

1 Department of Public Safety, Government of Brčko District of Bosnia and Herzegovina, Bulevara Mira 1, 76100 Brčko, Bosnia and Herzegovina
2 Institute of Agricultural Economics, Volgina 15, 11060 Belgrade, Serbia
3 College of Business Administration, American University in the Emirates, Dubai International Academic City, Dubai P.O. Box 503000, United Arab Emirates
4 Military Academy, University of Defence in Belgrade, Veljka Lukica Kurjaka 33, 11000 Belgrade, Serbia
* Correspondence: adispuska@yahoo.com

**Abstract:** This study deals with the selection of a sustainable supplier on the example of the agribusiness company Mamex from Bosnia and Herzegovina. The main problem of this research is the selection of a sustainable supplier as a part of the sustainable strategy of the Mamex company. One of the prerequisites is that suppliers must present sustainability principles in business by having an appropriate certificate. The results of the selection of sustainable suppliers are completed using a new hybrid fuzzy approach with the methods IMF SWARA (Improved Fuzzy Stepwise Weight Assessment Ratio Analysis) and fuzzy TRUST (multi-normalization multi-distance assessment) CRADIS (compromise ranking of alternatives from distance to ideal solution) methods. The innovative approach is reflected in the use of a combination of these methods, especially by combining the TRUST and CRADIS methods into one method. The IMF SWARA method shows that the most important main criterion is the economic criterion, while the least important is the social criterion. By applying the fuzzy TRUST CRADIS method, it is found that out of the observed six suppliers, the second supplier has the best indicators. These results are confirmed by other fuzzy methods: MABAC (multi-attributive border approximation area comparison), WASPAS (weighted aggregated sum product assessment), fuzzy SAW (simple additive weighting), MARCOS (measurement of alternatives and ranking according to compromise solution), ARAS (a new additive ratio assessment), and TOPSIS (technique for order preference by similarity to an ideal solution). This research shows that applying more normalization when ranking alternatives reduces the influence of individual normalizations, and this approach should be used in future research.

**Keywords:** sustainable supplier selection; agribusiness companies; fuzzy approach; IMF SWARA method; TRUST CRADIS method

## 1. Introduction

Companies cooperate with suppliers and customers to carry out their day-to-day operations. Raw materials and components needed for production are purchased from suppliers, while finished products are sold to customers. To adapt to the demands of customers, companies must form partnerships with suppliers who will assist them in this endeavor. The selection of suppliers is made using various criteria such as economic, social, ecological, technological, and others. To select the supplier that best meets these criteria, it is necessary to perform multi-criteria decision-making (MCDM). In addition, different methods of supplier selection are applied due to the demand of customers and the market, as well as the public.

Public concern for environmental issues has grown significantly in the past few years [1]. More and more efforts are being made to solve these problems. Through the operations of the company, efforts are being made to transform the business to gain a

competitive advantage in the international market. Companies are increasingly applying sustainability in business, especially in supply chain management. Sustainable supply chain management is gaining increasing attention in the scientific and business world [2]. A sustainable supply chain includes three key sustainability factors, namely the economic, environmental, and social factors in business. By applying the supply chain, companies perform all operations that connect them with suppliers and customers [3].

For the daily performance of business activities, any company must have enough raw materials and components that it procures from suppliers. The selection of suppliers is the first step in any production because it is first necessary to acquire raw materials and components to make the finished product [4]. To apply sustainability in business, it is necessary to include suppliers in those processes. Therefore, the selection of a sustainable supplier is a very important and challenging problem for every company [5]. This problem is further complicated if they are from the agribusiness sector. Customers are particularly concerned about food quality as more and more attention is paid to healthy living [6]. That is why agribusiness companies must choose a supplier who will supply them with safe raw materials so that the finished product is of high quality and healthy.

To reduce operating costs, agribusiness companies have completely closed their operations in a way that they produce raw materials, process them, and sell them as a finished product. Thus, agribusiness companies have developed operations from the farm to the plate that is, from the farm to the final consumer [7]. To be able to do this, companies have to invest in multiple activities, because they do not process the raw materials, they get from suppliers but produce them in-house. The only exceptions are seeds and fertilizers that they buy from suppliers. Therefore, it is necessary to select suppliers who will help them in these activities.

This paper aims to use the example of the company Mamex Bijeljina to select sustainable suppliers that would help them develop a sustainable business. Since, when choosing the suppliers, there are many criteria by which they are evaluated, and the company has many suppliers at its disposal, this business problem should be solved by applying multi-criteria decision-making (MCDM) [8]. There are a large number of MCDM methods available MCDM. For this study, the IMF SWARA (Improved Fuzzy Stepwise Weight Assessment Ratio Analysis) and TRUST CRADIS (Compromise Ranking of Alternatives from Distance to Ideal Solution) methods were used. These methods were used in the fuzzy set because they use linguistic values that are more adapted to human thinking [9]. The decision-makers (DM) in the Mamex Bijeljina Company evaluated the importance of the criteria and then selected suppliers with those criteria.

Based on the main research goal, additional research objectives are set to:

- Select a sustainable supplier using a hybrid fuzzy approach;
- Use an innovative method for ranking alternatives;
- Determine how the importance of the criteria affects the ranking of the selected suppliers.

In addition, this study also addresses certain research gaps. When applying the MCDM method, normalization has a great influence on the ranking of alternatives. Based on this, this study offers an innovative solution of using multiple normalizations, where all normalizations are considered and given equal weight. In this way, all the advantages of certain normalizations are used. Another drawback is related to the fact that when using the MCDM method, the steps of only one of the methods are used. This study shows that it is possible to use multiple steps of different methods with a single methodology to take advantage of those methods. The third drawback addressed by this study is that when implementing sustainability in business, all partners must contribute to it, which is why a sustainable selection of suppliers was considered because suppliers represent one of the company's key partners.

Section 2 reviews the literature on the application of MCDM methods in supplier selection. Section 3 describes the methodology and methods used in the paper. Section 4

presents the results and discussion. The conclusion, limitations, and directions for future research are given in Section 5.

## 2. Literature Review

The selection of suppliers is a decision that all companies face. Such decisions are complex, as they require the identification, consideration, and analysis of many factors [10]. Some of these factors are costs, price, delivery flows, pollution control, reputation... [11] Each company when choosing a supplier with which to cooperate, uses several criteria by which it evaluates them.

Due to the existence of multiple criteria, this decision-making problem is solved using MCDM methods. Each of the methods has its methodology and steps. The first step of each of the MCDM methods is data normalization with the decision matrix [12]. When selecting suppliers, this matrix is formed in such a way that all the suppliers under consideration are evaluated using the selected criteria. Data normalization using different MCDM methods affects the ranking of alternatives [13]. Most MCDM methods in their original form use only one normalization. There are rare methods such as TRUST that use multiple normalizations. Therefore, in this research, two MDCM methods were merged into one, i.e., TRUST and CRADIS, to reduce the influence of normalization in the selection of suppliers.

The goal of every company is to find the right supplier that best meets the goals of that company. However, this problem has become more complicated in modern economic conditions because it is possible to obtain raw materials from any part of the world within a reasonable time. Because of this, it is now possible to find a large number of suppliers, but it is necessary to determine which suppliers are the best for the company and to establish good partnership relations with them [14].

In the process of suppliers' selection, sustainability plays an important role in business, making the decision more complicated [15]. The selection of suppliers becomes even more complicated because it is necessary to choose the supplier that will best assist in maintaining the sustainability of that company. This is addressed using sustainable supplier selection criteria. The basic sustainability criteria are economic, ecological, and social criteria [16]. Sustainable suppliers have been analyzed in many studies, so in this study, only some of them are mentioned.

Zhou and Xu [17] used the fuzzy methods DEMATEL (decision-making trial and evaluation laboratory), ANP (analytical network process), and VIKOR (in Serbian: *višekriterijumsko kompromisno rangiranje*) for the selection of a sustainable supplier using the example of a production and distribution company. Ulutaş et al. [18] used the Grey WISP (weighted sum-product) and Grey BWM (best–worst method) methods to select a sustainable supplier. Matić et al., [19] used the FUCOM (Full consistency method) and COPRAS (complex proportional assessment) methods when selecting sustainable suppliers in the construction industry. Wang et al. [20] used the fuzzy AHP (analytic hierarchy process) and TOPSIS (technique for order preference by similarity to an ideal solution) methods to select sustainable suppliers in the example of the textile industry. Nedeljković [21] used multi-criteria decision-making with TOPSIS methods in an agricultural company when selecting sustainable suppliers.

Ecer and Pamučar [14] modified the CoCoSo (combined compromise solution) method to select a sustainable supplier for the production of household appliances. Memari et al. [22] used the TOPSIS method to perform the selection of a sustainable supplier on the example of a manufacturer of components for the automotive industry. Puška et al. [7] applied the PIPRECIA (pivot pairwise relative criteria importance assessment) and MABAC (multi-attributive border approximation area comparison) methods to select sustainable agricultural pharmacies for the procurement of planting and other materials for agricultural production. Nedeljković [15] investigates the selection of a sustainable supplier in a local agricultural company using fuzzy logic and methods DEMATEL (decision-making trial and evaluation laboratory). Liu et al. [23] used the ANP and VIKOR

(in Serbian: *višekriterijumsko kompromisno rangiranje*) methods to select the supplier that best contributes to achieving a sustainable supply chain.

As can be seen from this literature review, various MCDM methods have been used in the selection of sustainable suppliers. Based on this, it can be concluded that MCDM methods are used in the selection of a sustainable supplier.

## 3. Methodology

Mamex Bijeljina is one of the few companies applying the business philosophy of in-house processes from the farm to the consumer. They own the largest agricultural farm in Bosnia and Herzegovina for vegetable production. Based on this, they procure various raw materials from agricultural seeds, fertilizers, and irrigation systems, through packaging and raw materials. That is why they have a variety of suppliers with whom they enter into business cooperation. To compete in the market, they decided to apply sustainable production. The Mamex company has to change its business approach. In this endeavor, the company's partners must also assist them, mainly their suppliers and customers. To achieve sustainability in their business operations, raw materials and components must be procured from suppliers that will be aligned with sustainability principles. For achieving this, they need suppliers to support their aim.

Together with the Mamex company, the following phases were applied in this research:

- Phase 1. Initial phase
- Phase 2. Data collection
- Phase 3. Data processing and analysis
- Phase 4. Examination of results and sensitivity analysis

The first one was the initial phase (Figure 1). In this phase, together with the company, the suppliers were determined. In total, six alternative suppliers were considered in this decision-making process. These six suppliers represent agricultural pharmacies from which the company Mamex acquires all raw materials, including means for agro protection. Due to the specificity of raw materials and protective equipment, they must be harmless to health and the environment. Thus, these suppliers must apply sustainability in their business. To evaluate these suppliers and choose the supplier that best solves the defined decision-making problem, it was necessary to determine the criteria by which these suppliers should be evaluated. The evaluation of alternatives and criteria was carried out by three experts employed by the Mamex company. The CEO of the company was chosen as the first expert, the head of a production was chosen as the second expert, and the procurement director was chosen as the third expert. These experts were chosen because they are the most knowledgeable in the procurement system and raw material needs for the Mamex company. A sustainable selection of suppliers was used. This choice implies the use of three basic criteria: economic, ecological, and social criteria. These criteria are the basic criteria of sustainability. Each of these criteria was divided into auxiliary criteria. To facilitate decision-making, five auxiliary criteria were used. In this way, fifteen auxiliary and three main criteria were used (Table 1).

The second phase of research was the collection of research data. For this purpose, a questionnaire consisting of three parts was used. The first part included the criteria and their definitions. The second part of the survey questionnaire was used for evaluating the criteria, namely the main criteria and auxiliary criteria. Experts first determined the most important criteria to which they assigned a value of zero (0) and evaluated the other criteria about the previous best criterion. A scale of linguistic values of seven levels and the eighth level for equal significance was available to the experts (Table 2). This procedure was applied for three main criteria and five auxiliary criteria for each of these criteria. The third part of the survey questionnaire was used for evaluating suppliers according to the observed auxiliary criteria. Experts rated the suppliers based on a seven-level value scale (Table 3). Each supplier was rated on this scale for each auxiliary criterion.

**Table 1.** Criteria for the selection of sustainable suppliers.

| Id | Criteria | Definition | Sources |
|----|----------|------------|---------|
| C1 | Economic criterion | | |
| C11 | Expenses | Value of procurement costs | [9,24–26] |
| C12 | Quality | The degree of satisfaction of customer requirements by suppliers | [14,24,26] |
| C13 | Delivery on time | Deliveries of products at the agreed time | [9,14,24,26] |
| C14 | Technological capacities | Technological capacities available to the supplier | [8,9,24] |
| C15 | Innovativeness | Introducing new and improved products and services to customers | [8,25] |
| C2 | Ecological criterion | | |
| C21 | Green product | Procurement of environmentally acceptable products | [7,19,20] |
| C22 | Reverse logistics and recycling | Material reuse and waste reduction | [7,19,25,26] |
| C23 | Eco product design | Produced by ecological standards | [8,9,19] |
| C24 | Environmental management system | Application of ISO 14001 standard in the organization | [9,14,26] |
| C25 | Pollution control | Standards for reducing the harmful impact on the environment | [9,14,19,24] |
| C3 | Social criteria | | |
| C31 | Employee rights | Respect for workers' rights | [9,14,26] |
| C32 | Reputation | Opinions on the organization by interest groups | [8,14,25,27] |
| C33 | Information Sharing | Sharing all important information about the organization | [14,26,27] |
| C34 | Training and development of employees | Investment in employees | [14,19,25,27] |
| C35 | Safety and security at work | A developed system of safety and protection at work | [10,25–27] |

**Table 2.** Membership function in the IMF SWARA method.

| Linguistic Variable | Abbreviation | TFN Scale | | |
|---------------------|--------------|-----------|---|---|
| Absolutely less significant | ALS | 1 | 1 | 1 |
| Dominantly less significant | DLS | 1/2 | 2/3 | 1 |
| Much less significant | MLS | 2/5 | 1/2 | 2/3 |
| Really less significant | RLS | 1/3 | 2/5 | 1/2 |
| Less significant | LS | 2/7 | 1/3 | 2/5 |
| Moderately less significant | MDLS | 1/4 | 2/7 | 1/3 |
| Weakly less significant | WLS | 2/9 | 1/4 | 2/7 |
| Equal significant | ES | 0 | 0 | 0 |

**Table 3.** Membership function for evaluation of alternatives.

| Linguistic Variable | Triangular Fuzzy Number |
| --- | --- |
| Very bad (VB) | (0, 0, 1) |
| Bad (B) | (0, 1, 3) |
| Medium bad (MB) | (1, 3, 5) |
| Medium (M) | (3, 5, 7) |
| Medium good (MG) | (5, 7, 9) |
| High (G) | (7, 9, 10) |
| Very good (VG) | (9, 10, 10) |

**Phase 1. Initial phase**
- Determination of experts for the selection of a sustainable supplier
- Selection of suppliers as alternatives used in research
- Selection of criteria for the selection of a sustainable supplier

**Phase 2. Data collection**
- Development of the survey questionnaire
- Distribution of survey questionnaires to experts
- Data collection from experts
- Data processing for analysis

**Phase 3. Data processing and analysis**
- Data processing and preparation for analysis
- Determining the weights of the criteria through the IMF SWARA method
- Determination of ranking by the TRUST CRADIS method

**Phase 4. Examination of results and sensitivity analysis**
- Validation of research results
- Defining scenarios for sensitivity analysis
- Carrying out a sensitivity analysis

**Figure 1.** Research Methodology.

The third phase of the research was data processing and analysis. After the data were collected from the experts, they were processed and prepared for analysis. To obtain the weights of the criteria, the IMF SWARA method was used. This method used processed data from the second part of the survey questionnaire, where experts evaluated the weight of the criteria. The collected linguistic values were transformed using the value scale used by the IMF SWARA method (Table 2) [28]. After that, the steps of this method were carried out and the weights of the criteria were determined. These weights are necessary to be able to rank alternatives since all MCDM methods require criterion weights.

After the weights were calculated, the alternatives were ranked. The TRUST CRADIS method was used to rank the alternatives. This method is a combination of the TRUST and CRADIS methods. To rank the alternatives, it was necessary to use the weights of the criteria and the values of the alternatives. Since the alternative values are in the form of a linguistic scale, it was necessary to transform this scale into fuzzy numbers (Table 3) using the membership function to use the fuzzy TRUST CRADIS method. After that, the steps of the fuzzy TRUST CRADIS method were applied, and the alternatives are ranked.

The fourth stage was the examination of the results and the sensitivity analysis. When examining the results of the research, the validation of the results using other fuzzy MCDM methods was used. These methods used the obtained weights and values of the alternatives,

and they formed a ranking list of suppliers. This analysis aimed to confirm or refuse the results obtained using the fuzzy TRUST CRADIS method. After that, a sensitivity analysis was carried out, which aims to examine how the change in the weights of the criteria has an impact on the ranking of the alternatives. In this way, the influence of individual auxiliary criteria on the ranking of individual suppliers was examined.

This section may be divided into subheadings. It should provide a concise and precise description of the experimental results, their interpretation, as well as the experimental conclusions that can be drawn.

### 3.1. IMF SWARA Method

IMF SWARA represents a modification of the fuzzy SWARA method [28]. This method uses the same steps as the fuzzy SWARA method except that it applies a different value scale (Table 2). The other steps are the same as in the SWARA method:

Step 1. Identification and selection of criteria.

Step 2. Sorting the criteria according to their importance from the most important to the least important.

Step 3. Determining the relative importance of criteria. Here, the criterion that has the greatest significance takes on the value of zero (0), while the value of the other criteria is determined by their significance.

Step 4. Calculation of the coefficient value $k_j$, based on expression:

$$k_j = \begin{cases} 1 \; if \; j = 1 \\ s_j + 1 \; if \; j > 1 \end{cases} \tag{1}$$

Step 5. Calculation of significance values $q_j$, based on expression:

$$q_j = \begin{cases} 1 \; if \; j = 1 \\ \frac{q_j - 1}{k_j} \; if \; j > 1 \end{cases} \tag{2}$$

Step 6. Calculating the weight of criteria $w_j$, based on expression:

$$w_j = \frac{q_j}{\sum_{j=1}^{n} q_k} \tag{3}$$

### 3.2. TRUST CRADIS Method

The fuzzy TRUST CRADIS method is a combination of TRUST and CRADIS methods. The TRUST (a multi-normalization multi-distance assessment) method was developed by Torkayesh and Deveci [29]. This method is specific because it uses four normalizations, three linear normalizations, and a logarithmic normalization. This specificity was inserted into the CRADIS method to perform four normalizations instead of one. Other steps are applied from the fuzzy CRADIS method. A similar approach was taken in the comparative analysis of the global innovation index [30]. This approach is now extended to the fuzzy set. The CRADIS method was developed by Puška et al. [31], while the fuzzy approach was used by Puška et al. [32]. Based on this, this fuzzy TRUST CRADIS method has the following steps:

Step 1. Formation of the initial decision matrix. In this step, a decision-making matrix is formed based on the transformed linguistic values given by the experts for individual suppliers according to the observed criteria. The transformation of the linguistic matrix of decision-making is completed using the membership function (Table 3). In this way, the fuzzy decision matrix was formed and the sub-data for this matrix is developed $x_{ij} = x_{ij}^l, \; x_{ij}^m, \; x_{ij}^u$

Step 2. Normalization of the decision matrix. Since all criteria are in the form of benefit criteria, benefit normalization is used, which is performed by applying the following expressions:

$$\text{type} - 1 \text{ normalization}: \; n_{ij}^1 = \frac{x_{ij}^l}{\max x_j^u}, \frac{x_{ij}^m}{\max x_j^u}, \frac{x_{ij}^u}{\max x_j^u}, \tag{4}$$

where is max $x_j^u$ the largest value of the fuzzy number "u"

$$\text{type} - 2 \text{ normalization}: \; n_{ij}^2 = \frac{x_{ij}^l}{\sum_{i=1}^m x_{ij}^u}; \frac{x_{ij}^m}{\sum_{i=1}^m x_{ij}^u}; \frac{x_{ij}^u}{\sum_{i=1}^m x_{ij}^u} \tag{5}$$

where is $\sum_{i=1}^m x_{ij}^u$ the sum of alternative values for certain criteria for a fuzzy number "u"

$$\text{type} - 3 \text{ normalization}: \; n_{ij}^3 = \frac{(x_{ij}^l - \min\limits_j x_{ij}^l)}{(\max\limits_j x_{ij}^u - \min\limits_j x_{ij}^l)}; \frac{(x_{ij}^m - \min\limits_j x_{ij}^l)}{(\max\limits_j x_{ij}^u - \min\limits_j x_{ij}^l)}; \frac{(x_{ij}^u - \min\limits_j x_{ij}^l)}{(\max\limits_j x_{ij}^u - \min\limits_j x_{ij}^l)} \tag{6}$$

where is $\min\limits_j x_{ij}^l$ the smallest value of a fuzzy number "l", and max $x_j^u$ is the largest value of a fuzzy number "u"

$$\text{type} - 4 \text{ normalization}: \; n_{ij}^4 = \frac{\log(x_{ij}^l)}{\log(\prod_{i=1}^m x_{ij}^u)}; \frac{\log(x_{ij}^m)}{\log(\prod_{i=1}^m x_{ij}^u)}; \frac{\log(x_{ij}^u)}{\log(\prod_{i=1}^m x_{ij}^u)} \tag{7}$$

where is $\prod_{i=1}^m x_{ij}^u$ the product of alternative values for certain criteria for a fuzzy number "u".

After these normalizations are calculated, it reduces to a single value, while each normalization has the same importance.

$$n_{ij} = 0.25 \cdot n_{ij}^1 + 0.25 \cdot n_{ij}^2 + 0.25 \cdot n_{ij}^3 + 0.25 \cdot n_{ij}^4 \tag{8}$$

Step 3. Aggravation of decision matrix. Weighting is completed by multiplying the normalized value with the weight of that criterion. This is calculated using the following expression:

$$\widetilde{v}_{ij} = (v_{ij}^l, v_{ij}^m, v_{ij}^u) = \widetilde{n}_j \times \widetilde{w}_j \tag{9}$$

Step 4. Determination of ideal and anti-ideal values. After weighing the normalized data, the ideal and anti-ideal values of the matrix are calculated $v_{ij}$. The ideal value represents the highest value in the matrix $v_{ij}$ while the anti−ideal value represents the smallest value in the matrix $v_{ij}$.

$$t_i = \max\widetilde{v}_{ij}, \text{ where is } \widetilde{v}_{ij} = (v_{ij}^l, v_{ij}^m, v_{ij}^u) \tag{10}$$

$$t_{ai} = \min\widetilde{v}_{ij}, \text{ where is } \widetilde{v}_{ij} = (v_{ij}^l, v_{ij}^m, v_{ij}^u) \tag{11}$$

Step 5. Calculation of deviations from ideal and anti-ideal values. In this step, deviations from ideal and anti-ideal values of all data are calculated from the matrix $v_{ij}$.

$$d^+ = t_i - \widetilde{v}_{ij} \tag{12}$$

$$d^- = \widetilde{v}_{ij} - t_{ai} \tag{13}$$

Step 6. Formation of ideal and anti-ideal optimal alternatives about the deviation from the ideal and anti-ideal value. In this step, the smallest values of the criteria are searched for the ideal solution and the ideal solution of alternatives is formed. Next is the formation of the anti-ideal optimal alternative, which represents the maximum deviation of an individual alternative from the anti-ideal value for all criteria.

Step 7. Calculation of the sum of the deviations of individual alternatives from ideal and anti-ideal values.

$$s_i^+ = \sum_{j=1}^n d^+ \tag{14}$$



$$s_i^- = \sum_{j=1}^{n} d^- \tag{15}$$

Step 8. Defuzzification of scores of deviations of alternatives from ideal and anti-ideal solutions.

$$s_{i\ def}^{\pm} = \frac{d_i^l + 4d_i^m + d_i^u}{6} \tag{16}$$

Step 9. Calculation of the utility function for each alternative about the deviations from the optimal alternatives.

$$K_i^+ = \frac{s_0^+}{s_i^+} \tag{17}$$

$$K_i^- = \frac{s_i^-}{s_0^-} \tag{18}$$

where $s_0^+$ is the optimal ideal alternative, while je $s_0^-$ is the optimal anti-ideal alternative. The goal of each alternative is to be as close as possible to these ideal alternatives.

Step 10. Ranking of alternatives. The final value of the fuzzy CRADIS method is obtained by calculating the average deviation of the alternatives from the utility functions.

$$Q_i = \frac{K_i^+ + K_i^+}{2} \tag{19}$$

The ranking of alternatives is completed according to the obtained value of the fuzzy CRADIS method. The best alternative is the one with the highest value $Q_i$, while the worst is the one with the lowest value $Q_i$.

## 4. Results and Discussion

During the selection of suppliers, the Mamex company requested three experts to first evaluate the weight of the criteria and then the alternatives. When evaluating the criteria, they first evaluated the main criteria and then the auxiliary criteria. The experts first ranked these criteria and then determined to what extent each of the criteria is more important than the other in their opinion (Table 4). The first expert (DM1) believes that the most important is the economic criteria, while the ecological and social criteria, in his opinion, have the same importance. Experts two and three believe that economic and ecological criteria are the most important criteria. Based on these expert opinions, the economic criterion (C1) is the most important criterion when selecting a sustainable supplier, followed by the ecological criterion (C2), while in their opinion the least important is the social criterion (C3).

To determine the final weight of the criteria, the average weights obtained from these experts were taken (Table 5). In this way, all experts were given the same importance in decision-making, because their weights individually affect the final weights of the main criteria. The weights of auxiliary criteria were calculated in the same way.

After that, the experts ranked the auxiliary criteria and determined the values of how much certain criteria are better compared to the criteria that are ranked lower. They rated the ranked auxiliary criteria based on importance from their opinion (Table 2). Then, the steps of the IMF SWARA method are carried out for each expert and the average weight of all auxiliary criteria was calculated. Results were obtained showing that the most important auxiliary criterion for economic criteria (C1) is quality (C12), while the least important auxiliary criterion is technological capacities (C14). With the environmental criterion (C2), the most important auxiliary criterion based on the experts' opinion is pollution control (C25), while the least important auxiliary criterion is the environmental management system (C24). With the social criterion (C3), the most important auxiliary criteria according to experts' opinion are reputation (C32) and safety and security at work (C35), while the least important auxiliary criterion is employee training and development (C34) (Table 6). The final value of the weights was calculated by multiplying the weight of the main criteria with the weights of the auxiliary criteria.

**Table 4.** Evaluation of the main criteria and determination of weights.

| **DM1** | **s$_j$** | | | **k$_j$** | | | **q$_j$** | | | **w$_j$** | | |
|---|---|---|---|---|---|---|---|---|---|---|---|---|
| C1 | | | | 1.00 | 1.00 | 1.00 | 1.00 | 1.00 | 1.00 | 0.38 | 0.39 | 0.40 |
| C2 | 0.25 | 0.29 | 0.33 | 1.25 | 1.29 | 1.33 | 0.75 | 0.78 | 0.80 | 0.29 | 0.30 | 0.32 |
| C3 | 0.00 | 0.00 | 0.00 | 1.00 | 1.00 | 1.00 | 0.75 | 0.78 | 0.80 | 0.29 | 0.30 | 0.32 |
| | | | sum | 2.50 | 2.56 | 2.60 | | | | | | |
| **DM2** | **s$_j$** | | | **k$_j$** | | | **q$_j$** | | | **w$_j$** | | |
| C1 | | | | 1.00 | 1.00 | 1.00 | 1.00 | 1.00 | 1.00 | 0.35 | 0.36 | 0.36 |
| C2 | 0.00 | 0.00 | 0.00 | 1.00 | 1.00 | 1.00 | 1.00 | 1.00 | 1.00 | 0.35 | 0.36 | 0.36 |
| C3 | 0.22 | 0.25 | 0.29 | 1.22 | 1.25 | 1.29 | 0.78 | 0.80 | 0.82 | 0.28 | 0.29 | 0.29 |
| | | | sum | 2.78 | 2.80 | 2.82 | | | | | | |
| **DM3** | **s$_j$** | | | **k$_j$** | | | **q$_j$** | | | **w$_j$** | | |
| C1 | | | | 1.00 | 1.00 | 1.00 | 1.00 | 1.00 | 1.00 | 0.35 | 0.36 | 0.36 |
| C2 | 0.00 | 0.00 | 0.00 | 1.00 | 1.00 | 1.00 | 1.00 | 1.00 | 1.00 | 0.35 | 0.36 | 0.36 |
| C3 | 0.22 | 0.25 | 0.29 | 1.22 | 1.25 | 1.29 | 0.78 | 0.80 | 0.82 | 0.28 | 0.29 | 0.29 |
| | | | sum | 2.78 | 2.80 | 2.82 | | | | | | |

**Table 5.** Final weights of the main criteria.

| | **C1** | | | **C2** | | | **C3** | | |
|---|---|---|---|---|---|---|---|---|---|
| DM1 | 0.38 | 0.39 | 0.40 | 0.29 | 0.30 | 0.32 | 0.29 | 0.30 | 0.32 |
| DM2 | 0.35 | 0.36 | 0.36 | 0.35 | 0.36 | 0.36 | 0.28 | 0.29 | 0.29 |
| DM3 | 0.35 | 0.36 | 0.36 | 0.35 | 0.36 | 0.36 | 0.28 | 0.29 | 0.29 |
| Average | 0.36 | 0.37 | 0.37 | 0.33 | 0.34 | 0.35 | 0.28 | 0.29 | 0.30 |

**Table 6.** Weight values of auxiliary criteria.

| | **C11** | | | | **C12** | | | **C13** | | | **C14** | | | **C15** | | |
|---|---|---|---|---|---|---|---|---|---|---|---|---|---|---|---|---|
| DM1 | 0.21 | 0.23 | 0.24 | 0.27 | 0.28 | 0.29 | 0.16 | 0.18 | 0.19 | 0.12 | 0.14 | 0.16 | 0.16 | 0.18 | 0.19 |
| DM2 | 0.17 | 0.19 | 0.21 | 0.29 | 0.30 | 0.31 | 0.22 | 0.24 | 0.25 | 0.11 | 0.12 | 0.14 | 0.14 | 0.15 | 0.17 |
| DM3 | 0.20 | 0.21 | 0.22 | 0.25 | 0.26 | 0.27 | 0.20 | 0.21 | 0.22 | 0.15 | 0.16 | 0.18 | 0.15 | 0.16 | 0.18 |
| Average | 0.19 | 0.21 | 0.22 | 0.27 | 0.28 | 0.29 | 0.19 | 0.21 | 0.22 | 0.13 | 0.14 | 0.16 | 0.15 | 0.16 | 0.18 |
| | **C21** | | | | **C22** | | | **C23** | | | **C24** | | | **C25** | | |
| DM1 | 0.17 | 0.18 | 0.19 | 0.17 | 0.18 | 0.19 | 0.22 | 0.23 | 0.23 | 0.17 | 0.18 | 0.19 | 0.22 | 0.23 | 0.23 |
| DM2 | 0.19 | 0.20 | 0.21 | 0.19 | 0.20 | 0.21 | 0.24 | 0.25 | 0.26 | 0.14 | 0.15 | 0.17 | 0.19 | 0.20 | 0.21 |
| DM3 | 0.15 | 0.16 | 0.18 | 0.15 | 0.16 | 0.18 | 0.20 | 0.21 | 0.22 | 0.20 | 0.21 | 0.22 | 0.25 | 0.26 | 0.27 |
| Average | 0.17 | 0.18 | 0.19 | 0.17 | 0.18 | 0.19 | 0.22 | 0.23 | 0.23 | 0.17 | 0.18 | 0.19 | 0.22 | 0.23 | 0.24 |
| | **C31** | | | | **C32** | | | **C33** | | | **C34** | | | **C35** | | |
| DM1 | 0.22 | 0.23 | 0.23 | 0.22 | 0.23 | 0.23 | 0.17 | 0.18 | 0.19 | 0.13 | 0.14 | 0.15 | 0.22 | 0.23 | 0.23 |
| DM2 | 0.22 | 0.22 | 0.22 | 0.22 | 0.22 | 0.22 | 0.17 | 0.17 | 0.18 | 0.17 | 0.17 | 0.18 | 0.22 | 0.22 | 0.22 |
| DM3 | 0.19 | 0.20 | 0.21 | 0.24 | 0.25 | 0.25 | 0.15 | 0.16 | 0.17 | 0.15 | 0.16 | 0.17 | 0.24 | 0.25 | 0.25 |
| Average | 0.21 | 0.21 | 0.22 | 0.23 | 0.23 | 0.23 | 0.16 | 0.17 | 0.18 | 0.15 | 0.16 | 0.17 | 0.23 | 0.23 | 0.23 |

After the weights of the criteria were calculated, the suppliers were ranked. The experts evaluated each of the suppliers using auxiliary criteria (Table 3) so that each of the suppliers could get a value that ranged from very bad (VB) to very good (VG) (Table 7). To determine which of the suppliers has the best indicators, it was necessary to rank the suppliers according to their ratings. The first step was to transform the linguistic values into fuzzy numbers using the membership function (Table 3). After the linguistic values were transformed into fuzzy numbers, it was necessary to calculate the average values of the fuzzy numbers to obtain one fuzzy decision matrix. Using the average values of the fuzzy number, the same importance was given to all experts.

**Table 7.** Initial linguistic decision matrix.

| DM1 | C11 | C12 | C13 | C14 | C15 | C21 | C22 | C23 | C24 | C25 | C31 | C32 | C33 | C34 | C35 |
|-----|-----|-----|-----|-----|-----|-----|-----|-----|-----|-----|-----|-----|-----|-----|-----|
| S1 | MG | G | MG | M | MG | MG | M | G | MG | G | G | VG | MG | M | MG |
| S2 | MG | MG | MG | MG | G | MG | M | MG | MG | G | G | VG | MG | M | MG |
| S3 | G | VG | M | G | MG | M | M | MG | MG | G | G | G | G | M | M |
| S4 | G | M | MG | MG | MB | MB | MB | MG | M | M | VG | G | G | MB | M |
| S5 | M | MG | G | MG | M | B | M | M | M | MG | MG | MG | G | MB | MB |
| S6 | MB | G | MG | M | M | B | B | M | M | M | M | MG | M | MB | MB |

| DM2 | C11 | C12 | C13 | C14 | C15 | C16 | C17 | C21 | C22 | C23 | C24 | C25 | C31 | C32 | C33 |
|-----|-----|-----|-----|-----|-----|-----|-----|-----|-----|-----|-----|-----|-----|-----|-----|
| S1 | VG | G | VG | G | MG | G | MG | VG | G | MG | G | G | MG | MG | G |
| S2 | G | G | MG | G | MG | MG | MG | G | MG | MG | G | VG | G | G | G |
| S3 | G | MG | MG | MG | M | M | G | MG | MG | G | MG | G | G | VG | MG |
| S4 | G | MG | MG | MG | MB | MB | G | MB | MG | G | MG | MG | VG | G | VG |
| S5 | MG | M | M | M | MB | MB | MG | B | G | G | G | G | MG | G | G |
| S6 | M | M | MG | M | MB | M | MG | B | MG | MG | MG | MG | G | M | M |

| DM3 | C11 | C12 | C13 | C14 | C15 | C16 | C17 | C21 | C22 | C23 | C24 | C25 | C31 | C32 | C33 |
|-----|-----|-----|-----|-----|-----|-----|-----|-----|-----|-----|-----|-----|-----|-----|-----|
| S1 | G | G | G | MG | G | MG | G | MG | G | MG | MG | G | MG | MG | G |
| S2 | MG | VG | VG | MG | G | G | VG | G | G | MG | G | VG | M | MG | VG |
| S3 | MG | MG | G | MG | VG | VG | G | VG | MG | G | G | VG | M | VG | VG |
| S4 | M | MG | MG | M | MG | MG | G | MG | M | MG | MG | G | MB | G | MG |
| S5 | MB | MG | MG | G | MG | MG | M | MG | M | M | G | MG | MB | G | MG |
| S6 | B | MG | M | G | MG | MG | MB | MG | M | MG | MG | MG | B | MG | G |

After that, the steps of the MCDM method were carried out. The first step in each method was the normalization of the fuzzy decision matrix. In this step, one can find the specificity of the TRUST CRADIS method because four normalizations were applied to obtain the final normalized decision matrix. Each of these four normalizations was given the same importance to equally affect the supplier ranking. The normalization used can affect the ranking of alternatives, therefore this approach was applied to respect the normalizations used and to obtain the ranking of suppliers affected by all these normalizations.

Normalization type 1 performs normalization about the highest value of the criteria, so the values of this normalization tend to be one (1), so the highest spread of normalized data is precisely around the value of one (1). Normalization type 2 performs normalization based on the sum of the values of one criterion, which is why the values of this normalization are the highest compared to other normalizations and all values range up to the value of 0.20. Normalization type 3 performs normalization about the largest and smallest value, and the data values occupy values from zero (0) as the smallest value of the criterion to one as the largest value of the criterion. Normalization type 4 uses logarithms in the calculation, the

values of this normalization are similar to the values of normalization type 2. Their values are similar, and it can be said that when these normalizations are compared to each other, these two normalizations have approximate data values at most. Based on these specifics of individual normalization, the final value of the normalized data took on the specifics of these normalizations (Table 8).

**Table 8.** Normalization of data from the fuzzy decision matrix.

| Type 1 | | C11 | | | C12 | | | C13 | | ... | | C35 | |
|---|---|---|---|---|---|---|---|---|---|---|---|---|---|
| S1 | 0.72 | 0.90 | 1.00 | 0.70 | 0.90 | 1.00 | 0.72 | 0.90 | 1.00 | ... | 0.66 | 0.86 | 1.00 |
| S2 | 0.59 | 0.79 | 0.97 | 0.70 | 0.87 | 0.97 | 0.66 | 0.83 | 0.97 | ... | 0.72 | 0.90 | 1.00 |
| S3 | 0.66 | 0.86 | 1.00 | 0.63 | 0.80 | 0.93 | 0.52 | 0.72 | 0.90 | ... | 0.59 | 0.76 | 0.90 |
| S4 | 0.59 | 0.79 | 0.93 | 0.43 | 0.63 | 0.83 | 0.52 | 0.72 | 0.93 | ... | 0.59 | 0.76 | 0.90 |
| S5 | 0.31 | 0.52 | 0.72 | 0.43 | 0.63 | 0.83 | 0.52 | 0.72 | 0.90 | ... | 0.45 | 0.66 | 0.83 |
| S6 | 0.14 | 0.31 | 0.52 | 0.50 | 0.70 | 0.87 | 0.45 | 0.66 | 0.86 | ... | 0.38 | 0.59 | 0.76 |
| **Type 2** | | **C11** | | | **C12** | | | **C13** | | **...** | | **C35** | |
| S1 | 0.14 | 0.17 | 0.19 | 0.13 | 0.17 | 0.18 | 0.13 | 0.16 | 0.18 | ... | 0.12 | 0.16 | 0.19 |
| S2 | 0.11 | 0.15 | 0.19 | 0.13 | 0.16 | 0.18 | 0.12 | 0.15 | 0.17 | ... | 0.13 | 0.17 | 0.19 |
| S3 | 0.13 | 0.17 | 0.19 | 0.12 | 0.15 | 0.17 | 0.09 | 0.13 | 0.16 | ... | 0.11 | 0.14 | 0.17 |
| S4 | 0.11 | 0.15 | 0.18 | 0.08 | 0.12 | 0.15 | 0.09 | 0.13 | 0.17 | ... | 0.11 | 0.14 | 0.17 |
| S5 | 0.06 | 0.10 | 0.14 | 0.08 | 0.12 | 0.15 | 0.09 | 0.13 | 0.16 | ... | 0.08 | 0.12 | 0.15 |
| S6 | 0.03 | 0.06 | 0.10 | 0.09 | 0.13 | 0.16 | 0.08 | 0.12 | 0.16 | ... | 0.07 | 0.11 | 0.14 |
| **Type 3** | | **C11** | | | **C12** | | | **C13** | | **...** | | **C35** | |
| S1 | 0.68 | 0.88 | 1.00 | 0.47 | 0.82 | 1.00 | 0.50 | 0.81 | 1.00 | ... | 0.44 | 0.78 | 1.00 |
| S2 | 0.52 | 0.76 | 0.96 | 0.47 | 0.76 | 0.94 | 0.38 | 0.69 | 0.94 | ... | 0.56 | 0.83 | 1.00 |
| S3 | 0.60 | 0.84 | 1.00 | 0.35 | 0.65 | 0.88 | 0.13 | 0.50 | 0.81 | ... | 0.33 | 0.61 | 0.83 |
| S4 | 0.52 | 0.76 | 0.92 | 0.00 | 0.35 | 0.71 | 0.13 | 0.50 | 0.88 | ... | 0.33 | 0.61 | 0.83 |
| S5 | 0.20 | 0.44 | 0.68 | 0.00 | 0.35 | 0.71 | 0.13 | 0.50 | 0.81 | ... | 0.11 | 0.44 | 0.72 |
| S6 | 0.00 | 0.20 | 0.44 | 0.12 | 0.47 | 0.76 | 0.00 | 0.38 | 0.75 | ... | 0.00 | 0.33 | 0.61 |
| **Type 4** | | **C11** | | | **C12** | | | **C13** | | **...** | | **C35** | |
| S1 | 0.16 | 0.17 | 0.18 | 0.15 | 0.17 | 0.17 | 0.15 | 0.16 | 0.17 | ... | 0.14 | 0.16 | 0.18 |
| S2 | 0.14 | 0.16 | 0.18 | 0.15 | 0.16 | 0.17 | 0.14 | 0.16 | 0.17 | ... | 0.15 | 0.17 | 0.18 |
| S3 | 0.15 | 0.17 | 0.18 | 0.14 | 0.16 | 0.17 | 0.12 | 0.15 | 0.16 | ... | 0.13 | 0.15 | 0.17 |
| S4 | 0.14 | 0.16 | 0.18 | 0.11 | 0.14 | 0.16 | 0.12 | 0.15 | 0.17 | ... | 0.13 | 0.15 | 0.17 |
| S5 | 0.09 | 0.13 | 0.16 | 0.11 | 0.14 | 0.16 | 0.12 | 0.15 | 0.16 | ... | 0.11 | 0.14 | 0.16 |
| S6 | 0.02 | 0.09 | 0.13 | 0.12 | 0.15 | 0.16 | 0.11 | 0.14 | 0.16 | ... | 0.10 | 0.13 | 0.15 |
| **Final** | | **C11** | | | **C12** | | | **C13** | | **...** | | **C35** | |
| S1 | 0.43 | 0.53 | 0.59 | 0.36 | 0.51 | 0.59 | 0.38 | 0.51 | 0.59 | ... | 0.34 | 0.49 | 0.59 |
| S2 | 0.34 | 0.47 | 0.57 | 0.36 | 0.49 | 0.56 | 0.32 | 0.46 | 0.56 | ... | 0.39 | 0.52 | 0.59 |
| S3 | 0.38 | 0.51 | 0.59 | 0.31 | 0.44 | 0.54 | 0.21 | 0.38 | 0.51 | ... | 0.29 | 0.42 | 0.52 |
| S4 | 0.34 | 0.47 | 0.55 | 0.16 | 0.31 | 0.46 | 0.21 | 0.38 | 0.54 | ... | 0.29 | 0.42 | 0.52 |
| S5 | 0.16 | 0.30 | 0.43 | 0.16 | 0.31 | 0.46 | 0.21 | 0.38 | 0.51 | ... | 0.19 | 0.34 | 0.47 |
| S6 | 0.05 | 0.16 | 0.30 | 0.21 | 0.36 | 0.49 | 0.16 | 0.32 | 0.48 | ... | 0.14 | 0.29 | 0.42 |

After the fuzzy decision matrix was normalized, the next step of the fuzzy TRUST CRADIS method is to make this decision matrix aggravated. Here, the normalized data were multiplied with the weights of the sub-criteria (expression (9)). The next step was to find the ideal and anti-ideal values. The ideal value represents the largest value of the difficult decision matrix, while the anti-ideal value is the smallest value of the difficult decision matrix (expressions (10) and (11)). The deviation of all values of the difficult decision matrix from these ideal and anti-ideal values was then calculated (expressions (12) and (13)). The next step was to determine the optimal alternatives (S0), namely ideal and anti-ideal alternatives. The ideal alternative is the one that deviates the least from the ideal value, while the anti-ideal value is the one that deviates the most from the anti-ideal value. After that, the sum of these deviations was performed for the alternatives and the optimal alternatives (expressions (14) and (15)). Then, defuzzification of these sums was performed (expression (16)). The next step of this method was the calculation of the utility function (expressions (17) and (18)), the final value of the alternatives was calculated (expression (19)) and a ranking list of alternatives was formed (Table 9).

**Table 9.** Deviation of alternatives from optimal alternatives and final ranking.

|  | $s^+$ | $s^-$ | $Defs^+$ | $Defs^-$ | $K_i^-$ | $K_i^-$ | $Q_i$ | RANK |
|---|---|---|---|---|---|---|---|---|
| S1 | (0.22 0.32 0.34) | (0.26 0.29 0.29) | 0.304 | 0.286 | 0.894 | 0.899 | 0.897 | 2 |
| S2 | (0.22 0.31 0.34) | (0.27 0.29 0.29) | 0.302 | 0.288 | 0.900 | 0.905 | 0.902 | 1 |
| S3 | (0.23 0.33 0.36) | (0.25 0.27 0.27) | 0.320 | 0.270 | 0.850 | 0.848 | 0.849 | 3 |
| S4 | (0.33 0.42 0.43) | (0.16 0.18 0.20) | 0.408 | 0.181 | 0.666 | 0.571 | 0.619 | 4 |
| S5 | (0.36 0.46 0.45) | (0.13 0.15 0.18) | 0.439 | 0.151 | 0.620 | 0.475 | 0.548 | 5 |
| S6 | (0.41 0.51 0.50) | (0.08 0.10 0.13) | 0.490 | 0.099 | 0.555 | 0.313 | 0.434 | 6 |
| $S_0$ | (0.18 0.28 0.32) | (0.30 0.32 0.31) | 0.272 | 0.318 |  |  |  |  |

The results show that supplier S2 has the best indicators, followed by supplier S1, while supplier S6 has the worst indicators. Based on these results, the Mamex company should choose suppliers S2 and S1 for business cooperation because according to experts, they would best contribute to sustainability for this company. In this way, this company would be more competitive in the market.

To examine the obtained results, their validation was carried out. This procedure involved comparing the results obtained with several different methods [33–36]. If the results of this method differ from the results of other methods, the question arises whether these results are valid [37–39]. The validation was performed with six other methods, namely: fuzzy MABAC (multi-attributive border approximation area comparison) method, fuzzy MARCOS (measurement of alternatives and ranking according to compromise solution), fuzzy WASPAS (weighted aggregated sum product assessment), fuzzy SAW (simple additive weighting), fuzzy ARAS (a new additive ratio assessment) and fuzzy TOPSIS (a technique for order preference by similarity to ideal solution). The specificity of these methods is that different normalizations are used in these methods. For example, the fuzzy MABAC (multi-attributive border approximation area comparison) method uses normalization type 3, fuzzy ARAS (a new additive ratio assessment) normalization type 2, and fuzzy MARCOS (measurement of alternatives and ranking according to compromise solution) normalization type 1. In this way, the ranking order of suppliers was examined by applying different normalizations.

As can be seen from the validation of the results, the deviation from the results exists only when applying the fuzzy TOPSIS method (Figure 2). In this method, suppliers S2 and S1 switched places, so supplier S1 was first in the ranking list and supplier S2 was second in the ranking list. The reason for this difference should be sought in the way in which the fuzzy TOPSIS method is implemented. It uses the same normalization as the fuzzy MARCOS (measurement of alternatives and ranking according to compromise solution) or fuzzy

WASPAS (weighted aggregated sum product assessment), and fuzzy SAW (simple additive weighting) methods, but this difference occurs when calculating data deviations. Based on these validation results, it can be concluded that the fuzzy TRUST CRADIS method does not differ from other methods and can be used when solving MCDM problems.

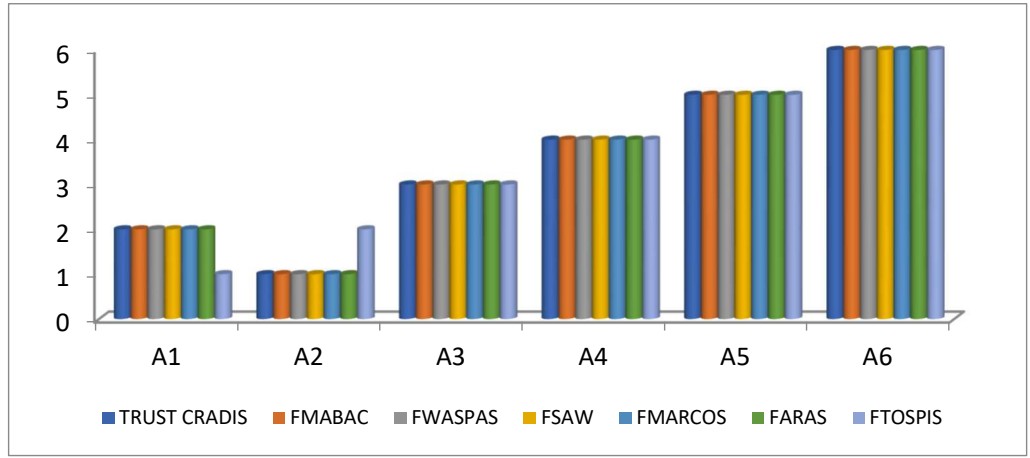

**Figure 2.** Validation of results.

After the validation was completed, a sensitivity analysis was performed. This analysis aims to examine how changes in the weight of one auxiliary criterion affect the ranking of alternatives [40–42]. In this case, the weight of one of the auxiliary criteria was changed by 30, 60, and 90%, so it was observed how this change in the weight of the criteria affects the ranking of the alternatives. Since there are 15 auxiliary criteria, 45 scenarios were executed. The results of this analysis show that there is a difference only in the ranking of suppliers S2 and S1, the other rankings do not change. This change is for six scenarios and auxiliary criteria C14, C15, C31, C32, and C34. With these auxiliary criteria, supplier S2 has better indicators, and when the importance of these criteria was reduced, supplier S1 is ranked better (Figure 3). Based on this, it can be found that supplier S2 is sensitive to changes in the weight of these criteria.

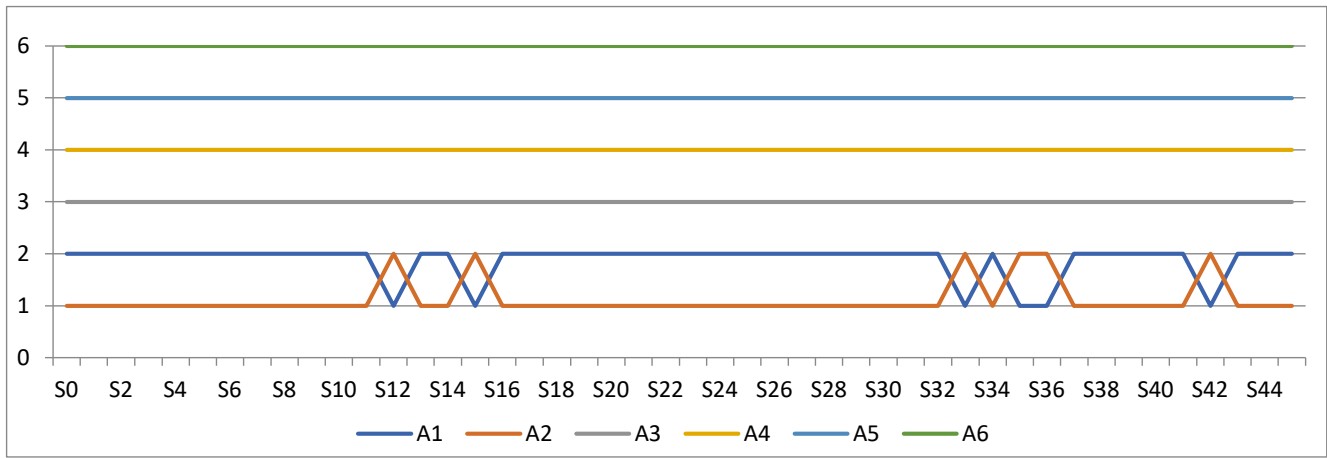

**Figure 3.** Sensitivity analysis.

All the conducted analyses show that supplier S2 has better indicators than supplier S1. Thus, if the Mamex company has to decide on one supplier, it should be supplier S2, and if it can choose more suppliers, this company should also establish partnership relations with supplier S1.

To cope with the growing market demand for a product that implements the principles of food safety and environmental protection, agribusiness companies must apply

sustainability in their operations. Sustainability is based on economic, environmental, and social factors [43]. By applying the principles of sustainability, the competitiveness of these companies is improved. Resilience must be considered in all company processes to cope with these market demands, especially in supply chain applications. The selection of suppliers is the first step and the first problem within the supply chain [44]. This problem is one of the key problems of every company [45]. The reason for this should be sought in the fact that suppliers assist in the realization of sustainability goals [46].

To operate sustainably, every company must select sustainable suppliers. The supplier is the one who serves the company with quality raw materials and components on time that does not harm the environment and human health [19]. The Mamex company is one of the companies that has a closed production cycle including activities, agricultural production, harvesting, processing, preparation, and making finished products for customers. This company also owns the largest agricultural farm in Bosnia and Herzegovina, where they produce various types of vegetables. These vegetables are raw materials for the production of finished products.

To apply sustainability in business, the Mamex company was focused to select a sustainable supplier. For this purpose, three experts were appointed who determined the criteria. Since a sustainable selection of suppliers was applied, three main criteria of sustainability were used: economic, ecological, and social criteria [14]. These criteria were divided into five more auxiliary criteria. It was chosen to be the same number of auxiliary criteria for each sustainable criterion in order not to give more importance to one of these main criteria. Since in this problem, several alternatives were observed according to several criteria, this decision-making problem belongs to the classic MCDM problem [47]. This decision problem was solved using different MCDM methods.

This research used a methodology based on the IMF SWARA and TRUST CRADIS methods. The IMF SWARA method was used to determine the importance of the criteria by determining the weights of the main and auxiliary criteria. Unlike the classic SWARA, which was developed by Keršuliena et al. [48], the IMF SWARA method uses an adapted value scale because the classical scale did not provide good results. It happened that although there were better values, certain criteria had less weight, and criteria with the same importance had different weights [28]. For these reasons, this version of the SWARA method was used.

Using the IMF SWARA method in determining the value of the weights, the findings show that the economic factor is more important to the experts than other factors. Within the framework of economic auxiliary criteria, the most important auxiliary criterion is quality. Therefore, for experts, it is most important to have quality raw materials and components, so that their final product is as high quality as possible. A good product cannot be made from faulty components and raw materials. Therefore, for the Mamex company it is most important to be supplied with quality raw materials and components.

Observing the auxiliary criteria for the environmental criterion, for experts, the most important is pollution control. Selecting this auxiliary criterion as the most important means that the supplier has implemented standards to reduce environmental impact. With the application of these standards, the raw materials and components should have no harmful impact on the environment. With the social criterion, the most important auxiliary criteria are reputation and safety and security at work. Thus, each company must have a good reputation from the interest groups, because this is important information for entering into business cooperation with other companies. If the company has a bad reputation, it will be avoided by other companies and customers themselves [49]. Therefore, every company must apply good business practices to make its reputation as good as possible [50]. In addition, the company must apply certain standards to protect the health and safety of employees. These two auxiliary criteria are the most important when choosing a supplier.

When evaluating the criteria and alternatives in this research, linguistic values were used. These values are closer to human thinking than classical ratings [51]. Sometimes it is easier to choose qualitative rating values such as good and bad rather than to evaluate

them quantitatively using numerical ratings. To work with these linguistic values, a fuzzy approach was used [52]. In this research, an innovative approach based on a combination of two methods, namely TRUST and CRADIS, was used. The approach of applying four normalizations was taken from the TRUST method, while the other steps were taken from the CRADIS method. This is because normalization has an impact on the final ranking of the alternatives [53]. To reduce the influence of individual normalizations, four normalizations were used in this approach.

The result of this approach showed that supplier S2 has the best sustainability indicators. It should be mentioned that supplier S1 also has good indicators even when using the fuzzy TOPSIS method in the validation of the results. The validation of the results showed that the results obtained with the fuzzy TRUST CRADIS method are the same as with other fuzzy methods, only with the fuzzy TOPSIS method there are differences. In this way, it has been proven that this approach provides the same or similar results and can be used in other MCDM problems. In addition to the validation of the results, a sensitivity analysis was also performed. This analysis showed that there was only a change in the ranking of suppliers S2 and S1. This analysis showed that suppliers S2 and then S1 are the most sustainable suppliers for the Mamex company, and they should establish cooperation with them, while they should not have business cooperation with supplier S6 since this supplier showed the worst results.

## 5. Conclusions

In this study, the selection of sustainable suppliers was used to support the business sustainability aim of the Mamex company from Bosnia and Herzegovina. This company has its production of vegetables and final products with a closed production process. Due to the specifics of the production of food products, this company must offer a healthy and safe product. Thus, they have decided to apply sustainability in business. In doing so, suppliers play a vital role, and they also need to apply sustainability in their business. To select suppliers that will best assist in this aim, sustainability criteria were used, which were divided into five auxiliary criteria. Six suppliers were evaluated with these criteria.

Based on the opinion of experts, the results of this study were obtained. The results of the research for the weights of the criteria showed that the most important were the economic and then the environmental criteria, while the least important was the social criteria when selecting sustainable suppliers. Within the economic criteria, the most important was the quality criterion, while within the ecological it was pollution control. This was obtained using the IMF SWARA method. When ranking suppliers' innovative fuzzy TRUST CRADIS method, results were obtained showing that supplier S2 has the best indicators and is the first choice when selecting a sustainable supplier. These results were confirmed in the validation of the results and with the sensitivity analysis.

Only three experts participated in this study, which can be considered the limitations of this study. Mamex is a newly established company with a limited number of employees who could be considered experts in this field. That is why they suggested these three experts who deal with procurement in this company. However, the qualification of experts and not their number was important for this study. Those experts who have the most important role in decision-making in this company participated in this study. In addition, the disadvantage of this study is that five auxiliary criteria were taken in each of the main criteria. However, increasing the number of criteria would only complicate the decision-making process, because experts would have to evaluate alternatives with more criteria. The selection of these criteria was made in cooperation with experts, and they considered these criteria as the most important in the selection of a sustainable supplier for the Mamex company.

This study contributes to the existing literature with the application of the TRUST CRADIS method, aiming for stability in decision-making. It neutralizes the differences between normalizations because four normalizations were used simultaneously. Applying this approach, the advantage of all these normalizations was taken. In addition, the

application of this method represents an innovation in the field of MCDM because it combines two methods into one. This approach opens up the possibility of creating new hybrid methods that could use already existing methods. In this way, a new segment of MCDM methods is opened. This methodology contributes to more consistent rankings as shown by the sensitivity analysis. In addition, the connection with other fuzzy methods is also better, as shown by the validation of the results.

**Author Contributions:** Conceptualization, A.P. and M.N.; methodology, A.P. and D.B.; software, A.P.; validation, M.N., I.S. and D.B.; formal analysis, A.P. and D.B.; investigation, M.N.; resources, M.N.; data curation, D.B.; writing—original draft preparation, A.P.; writing—review and editing, A.P.; visualization, D.B.; supervision, I.S. and D.B.; project administration, M.N.; funding acquisition, I.S. All authors have read and agreed to the published version of the manuscript.

**Funding:** This research received no external funding.

**Institutional Review Board Statement:** Not applicable.

**Informed Consent Statement:** Not applicable.

**Data Availability Statement:** Not applicable.

**Conflicts of Interest:** The authors declare no conflict of interest.

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
