# Peer review of "Application of Fuzzy TRUST CRADIS Method for Selection of Sustainable Suppliers in Agribusiness"

_sustainability, doi:10.3390/su15032578_

Round 1
Reviewer 1 Report
Dear respected authors,
1. The Abstract section should be rechecked. The first sentence needs to be supported by a reference, hence it is suggested to be placed in the Introduction section, or be rewritten as “one of the main issues in production …”. In line 17, it should be mentioned that whether proving “sustainability in business” needs a certificate to be taken by the companies or the respected authors state it in general. In lines 25 and 26, the supplies S2 in meaningless for the authors as there is no explanation about it before mentioning it. The number of supplies that have been considered in this study should be mentioned in this section. In line 26, the other fuzzy methods should be named.
2. It is recommended to modify the title of the manuscript to “Application of Innovation Fuzzy Decision-Making Methods in the Selection of a Sustainable Supplier in Agribusiness” to contain more information about the study.
3. Considering the mentioned aims in lines 72 to 75, proposing a new fuzzy MCDM approach should be mentioned in the Abstract section.
4. In lines 87 and 88, the phrase “many factors” has been in two consecutive sentences which need to be referred to some references and/or at least the main ones should be named.
5. The research gap, that this study wants to fill, should be highlighted in the Literature review section.
6. Referencing should be corrected in Table 1. For instance, the first row should be [9,19-22], etc.
7. Everywhere in the text that needs to mention any table, the first letter should be in capital form. For instance, “table 1” in line 153 should be “Table 1”. The text should be checked and other similar mistakes should be corrected.
8. In line 144, it has been mentioned that six alternative suppliers have been considered in this study, but there is no more information about these suppliers.
9. In line 426, the positions and organizational level of the selected experts should be explained. In addition, it is recommended to mention how these experts have been selected.
10. The text needs a major grammatical check. For instance, “On the example …” in line 15 is grammatically incorrect, and should be modified.
Author Response
Reviewer 1.
Thank you very much for the suggestions to make the paper as good as possible. We have carefully considered all your suggestions in this paper.
Dear respected authors,
- The Abstract section should be rechecked. The first sentence needs to be supported by a reference, hence it is suggested to be placed in the Introduction section, or be rewritten as “one of the main issues in production …”. In line 17, it should be mentioned that whether proving “sustainability in business” needs a certificate to be taken by the companies or the respected authors state it in general. In lines 25 and 26, the supplies S2 in meaningless for the authors as there is no explanation about it before mentioning it. The number of supplies that have been considered in this study should be mentioned in this section. In line 26, the other fuzzy methods should be named.
All these suggestions have been accepted and this text has been corrected in the paper.
- It is recommended to modify the title of the manuscript to “Application of Innovation Fuzzy Decision-Making Methods in the Selection of a Sustainable Supplier in Agribusiness” to contain more information about the study.
The title of the paper has been corrected, but we had to drop Innovation because another reviewer requested it.
- Considering the mentioned aims in lines 72 to 75, proposing a new fuzzy MCDM approach should be mentioned in the Abstract section.
The abstract has been corrected in accordance with these suggestions
- In lines 87 and 88, the phrase “many factors” has been in two consecutive sentences which need to be referred to some references and/or at least the main ones should be named.
It has been corrected.
- The research gap, that this study wants to fill, should be highlighted in the Literature review section.
It has been added to the literature review
- Referencing should be corrected in Table 1. For instance, the first row should be [9,19-22], etc.
References in Table 1 have been corrected.
- Everywhere in the text that needs to mention any table, the first letter should be in capital form. For instance, “table 1” in line 153 should be “Table 1”. The text should be checked and other similar mistakes should be corrected.
This has been corrected throughout the text
- In line 144, it has been mentioned that six alternative suppliers have been considered in this study, but there is no more information about these suppliers.
The suppliers involved in this research are explained.
- In line 426, the positions and organizational level of the selected experts should be explained. In addition, it is recommended to mention how these experts have been selected.
The methodology explains why these experts were taken.
- The text needs a major grammatical check. For instance, “On the example …” in line 15 is grammatically incorrect, and should be modified.
The entire text has been proofread.
Reviewer 2 Report
sustainability-2160342-peer-review-v1
Application of Innovation Fuzzy Methods in the Selection of a Sustainable Supplier in Agribusiness.
In this paper, author shares a review study on the fuzzy methods in the selection of a sustainable supplier in agribusiness. The topic is interesting. However, the paper needs some corrections to meet expectations of a journal. Some of my concerns are as follows:
1. The literature review is too general and thus can’t indicate any novelty of the current study.
2. I do not agree with the title. The authors did not actually perform any studies on Innovative fuzzy method. Please revise the paper title.
3. Rewrite the abstract in simple present tense, and answer the following questions: What problem did you study and why is it important? What methods did you use? What were your main results?
4. In introduction, before starting the mentioned references, there is a need to add 8-9 lines related to the subject of the paper and write in general introduction. After that you should connect them with the references.
5. References are not in proper format. For instance, see reference numbers [25], [42], etc.
6. The last paragraph outlining the sections of the manuscript should be revised.
7. Page 5, Lines 179-185: Revise the paragraph by rewriting in simple present tense. Resolve the similar issues in all manuscript.
8. A graphic flowchart of the proposed approach to the problem should be provided.
9. In tables 4, 5, and 6, the calculations must be clarified.
10. The explanation of equations in section 3.2 is very confusing. Some parameters/variables are not clearly explained or not defined at all. And in some cases, there is explanation for stuff that do not occur in the corresponding equation. Please resolve this discrepancy.
11. Do not use so many abbreviations in the title, abstract and elsewhere. In the absence of stringent space constraints, the use of abbreviation is not a good idea because it decreases ease of reading if a person has to remember all the abbreviations. The paper is at places practically unreadable due to the excessive use of abbreviations.
12. And an abbreviation is used only if the term appears at least five times in the main text. (The Abstract, conclusion, figures, and tables don’t count.) If the term or phrase is used only two, three or four times it should not be abbreviated (The Chicago manual of style).
13. I also recommend the authors to professionally get the paper proofread, as I have noticed sentences with typos and inappropriate choice of words.
***
Author Response
Reviewer 2.
Thank you very much for the suggestions to make the paper as good as possible. We have carefully considered all your suggestions in this paper.
In this paper, the author shares a review study on the fuzzy methods in the selection of a sustainable supplier in agribusiness. The topic is interesting. However, the paper needs some corrections to meet the expectations of a journal. Some of my concerns are as follows:
- The literature review is too general and thus can’t indicate any novelty of the current study.
The literature review has been revised to highlight the gaps and contribution of the paper
- I do not agree with the title. The authors did not actually perform any studies on the Innovative fuzzy method. Please revise the paper title.
The title of the paper has been corrected according to the first reviewer's request and is now hopefully acceptable
- Rewrite the abstract in simple present tense, and answer the following questions: What problem did you study and why is it important? What methods did you use? What were your main results?
The summary has been corrected according to your requirements
- In introduction, before starting the mentioned references, there is a need to add 8-9 lines related to the subject of the paper and write in general introduction. After that you should connect them with the references.
The requested additional part of the text in the introduction has been written
- References are not in proper format. For instance, see reference numbers [25], [42], etc.
Those errors in the text were found and have been corrected
- The last paragraph outlining the sections of the manuscript should be revised.
This paragraph has been corrected
- Page 5, Lines 179-185: Revise the paragraph by rewriting in simple present tense. Resolve the similar issues in all manuscript.
These parts of the text have been corrected.
- A graphic flowchart of the proposed approach to the problem should be provided.
A diagram of how the methodology was done in this paper is drawn
- In tables 4, 5, and 6, the calculations must be clarified.
How these tables are formed has been explained in more detail
- The explanation of equations in section 3.2 is very confusing. Some parameters/variables are not clearly explained or not defined at all. And in some cases, there is explanation for stuff that do not occur in the corresponding equation. Please resolve this discrepancy.
The equations has been explained in more detail in section 3.2
- Do not use so many abbreviations in the title, abstract and elsewhere. In the absence of stringent space constraints, the use of abbreviation is not a good idea because it decreases ease of reading if a person has to remember all the abbreviations. The paper is at places practically unreadable due to the excessive use of abbreviations.
Efforts have been made to correct this in the paper
- And an abbreviation is used only if the term appears at least five times in the main text. (The Abstract, conclusion, figures, and tables don’t count.) If the term or phrase is used only two, three or four times it should not be abbreviated (The Chicago manual of style).
Efforts have been made to correct this in the paper
- I also recommend the authors to professionally get the paper proofread, as I have noticed sentences with typos and inappropriate choice of words.
The entire text has been proofread.
Round 2
Reviewer 1 Report
The methodology, procedures, and steps of the research have been modified and extended logically, and all the comments and suggestions of the reviewer have been met. With appreciation to the respected authors, and according to the reviewer's point of view, the manuscript is worth being published in the respected journal.
Author Response
Greetings,
Thank you for accepting our paper.
All the best.
Reviewer 2 Report
sustainability-2160342-peer-review-v2
Application of Innovation Fuzzy Methods in the Selection of a Sustainable Supplier in Agribusiness.
The manuscript looks much better than the previous version. However, there are still shortcomings. The article can be considered only if it is revised.
Comments and Suggestions for Authors:
1. In abstract, many abbreviations used without declaration, for example TRUST, MABAC, WASPAS, SAW, MARCOS, ARAS, and TOPSIS. Abbreviated terms must be fully defined first, and then the acronyms are used.
2. And an abbreviation is used only if the term appears at least five times in the main text. (The Abstract, conclusion, figures, and tables don’t count.) If the term or phrase is used only two, three or four times it should not be abbreviated.
3. The authors discussed several times what they are going to propose but did not discuss the research gap or what are the sole reasons for this research.
4. Line 560, conclusion section: The last paragraph is confusing and the future study issues are not promising. It doesn't clearly state what the scope of the study is.
5. The key point of this model is not prominent enough, please explain the algorithm design part and algorithm comparison part in detail.
6. Paper title has some grammar error and it should be revised. Why the word “in” appears two times in title.
7. There are still too many grammatical issues in the article. Many expressions and languages used are inappropriate. A native English speaker must proofread the manuscript,
8. The technical content of the manuscript is still not definitely sufficient to justify the findings.
***
Author Response
Greetings,
Thank you for your suggestions
- In abstract, many abbreviations used without declaration, for example TRUST, MABAC, WASPAS, SAW, MARCOS, ARAS, and TOPSIS. Abbreviated terms must be fully defined first, and then the acronyms are used.
All abbreviations in the paper have been corrected according to your guidelines
- And an abbreviation is used only if the term appears at least five times in the main text. (The Abstract, conclusion, figures, and tables don’t count.) If the term or phrase is used only two, three or four times it should not be abbreviated.
All abbreviations in the paper have been corrected according to your guidelines
- The authors discussed several times what they are going to propose but did not discuss the research gap or what are the sole reasons for this research.
In the introduction of the paper, the gaps in the research are listed
- Line 560, conclusion section: The last paragraph is confusing and the future study issues are not promising. It doesn't clearly state what the scope of the study is.
The conclusion has been corrected according to your suggestions
- The key point of this model is not prominent enough, please explain the algorithm design part and algorithm comparison part in detail.
The key reasons for choosing a sustainable supplier are listed in the methodology
- Paper title has some grammar error and it should be revised. Why the word “in” appears two times in title.
The title of the paper has been corrected
- There are still too many grammatical issues in the article. Many expressions and languages used are inappropriate. A native English speaker must proofread the manuscript,.
The paper has been proofread
- The technical content of the manuscript is still not definitely sufficient to justify the findings.
Thank you for your valuable feedback. We hope that after we have corrected the paper it is acceptable.
Round 3
Reviewer 2 Report
sustainability-2160342-peer-review-v3
Application of Innovation Fuzzy Methods in the Selection of a Sustainable Supplier in Agribusiness.
Authors have made a correction in the paper if compared to the first submission. But unfortunately, there are recommendations to do before accepting the paper in this format.
1. There is some grammar mistake in paper title. It has to be clarify that reader can understand better.
2. In abstract, some sentences are lengthy, and are written in past tense, for example, “The IMF SWARA method showed that the most important main criterion”. The entire abstract should be in present tense. The author is suggested to revise all lengthy sentences available in abstract as well as in entire paper.
3. The last paragraph outlining the sections of the manuscript is difficult to understand. Revise it carefully; otherwise paper reject.
4. Line 181, the Figure 1 is not cited in text.
5. Merge section 4 & 5 to a single section, “Results and Discussion”.
6. The CONCLUSION section can be improved. Authors should avoid marginal explanations. They need to focus on contribution and then on their achievements. It is also better to write the numerical values of the improvements. Verbs must also be in the past tense.
7. There are a lot of formatting problems in references (i.e., References [49, 50, 51]. Please resolve the similar issues at other places also.
8. There are some inappropriate citations, for example, not supporting the claim being made or too many citations to the authors' own articles? This may cause the rejection of this paper. For example, the Reference numbers [3], [8], [9], [18], [30], [31], [32], [45], and [51].
***
Author Response
Authors have made a correction in the paper if compared to the first submission. But unfortunately, there are recommendations to do before accepting the paper in this format.
- There is some grammar mistake in paper title. It has to be clarify that reader can understand better.
The title of the paper has been corrected
- In abstract, some sentences are lengthy, and are written in past tense, for example, “The IMF SWARA method showed that the most important main criterion”. The entire abstract should be in present tense. The author is suggested to revise all lengthy sentences available in abstract as well as in entire paper.
The abstract has been corrected
- The last paragraph outlining the sections of the manuscript is difficult to understand. Revise it carefully; otherwise paper reject.
The last paragraph has been corrected in accordance with the papers published in the journal Sustainability
- Line 181, the Figure 1 is not cited in text.
Figure 1 has been inserted into the text of the paper
- Merge section 4 & 5 to a single section, “Results and Discussion”.
These sections in the paper have been merged
- The CONCLUSION section can be improved. Authors should avoid marginal explanations. They need to focus on contribution and then on their achievements. It is also better to write the numerical values of the improvements. Verbs must also be in the past tense.
The conclusion has been corrected and the contributions of this paper have been made
- There are a lot of formatting problems in references (i.e., References [49, 50, 51]. Please resolve the similar issues at other places also.
These references have been corrected and I would like to ask you to indicate what else we should possibly correct since we have included them before in the same way and there have been no comments so far.
- There are some inappropriate citations, for example, not supporting the claim being made or too many citations to the authors' own articles? This may cause the rejection of this paper. For example, the Reference numbers [3], [8], [9], [18], [30], [31], [32], [45], and [51].
Four references were removed from the paper and replaced by other references. Only those that are 100% related to this paper are included and should remain. In fact, I invented and introduced the CRADIS method.